# Hydraulic Features of Flow through Local Non-Submerged Rigid Vegetation in the Y-Shaped Confluence Channel

**Xuneng Tong, Xiaodong Liu \*, Ting Yang, Zulin Hua, Zian Wang, Jingjing Liu and Ruoshui Li** 

Key Laboratory of Integrated Regulation and Resource Development on Shallow Lake of Ministry of Education, 1 Xikang Road, Nanjing 210098, China; txn@hhu.edu.cn (X.T.); yang92005006@163.com (T.Y.); Zulinhua@hhu.edu.cn (Z.H.); wangzian1996@hotmail.com (Z.W.); 15205168389@163.com (J.L.); 1514050118@hhu.edu.cn (R.L.)
\* Correspondence: xdliu@hhu.edu.cn

**Abstract:** A laboratory measurement with acoustic Doppler velocimeter (ADV) was used to investigate the flow through a Y-shaped confluence channel partially covered with rigid vegetation on its inner bank. In this study, the flow velocities in cases with and without vegetation were measured by the ADV in a Y-shaped confluence channel. The results clearly showed that the existence of non-submerged rigid plants has changed the internal flow structure. The velocity in the non-vegetated area is greater than in the vegetated area. There is a large exchange of mass and momentum between the vegetated and non-vegetated areas. In addition, due to the presence of vegetation, the high-velocity area moved rapidly to the middle of the non-vegetated area in the vicinity of tributaries, and the secondary flow phenomenon disappeared. The presence of vegetation made the flow in non-vegetated areas more intense. The turbulent kinetic energy of the non-vegetated area was smaller than that of the vegetated area.

**Keywords:** Y-shaped confluence channel; non-submerged rigid vegetation; longitudinal velocity; secondary flow; turbulent kinetic energy

## 1. Introduction

Channel confluences are a common occurrence in fluvial networks, where significant changes can occur in hydraulics, sediment transport, water environment, and ecology. Many previous studies have concentrated on hydraulic characteristics in channel confluence areas [1–6]. Lan [7] divided the confluences into two types: (1) the confluence where post-confluence channel forms a linear extension of the upstream main channel (also named asymmetrical confluences), (2) the Y-shaped junction. According to Lan's data the quantities of two types of confluences are about equal in nature. Most previous work has paid more attention to the asymmetrical river confluences, but few studies have been conducted on the Y-shaped confluences [8–13]. Many studies confirmed that there are many differences between the two types of confluence channel. Nadia el al. [14], with their experimental work, found that the higher the junction angle, the wider and longer the retardation zone at the upstream junction corner and the separation zone, and the greater the flow deflection at the entrance of the tributary into the post-confluence channel.

The aquatic vegetation is ubiquitous in natural rivers. The vegetation has many ecological, aesthetic and economic benefits, such as providing a terrestrial wildlife habitat; improving water quality; stabilizing streambanks and floodplains; supplying energy subsidies for aquatic and terrestrial ecosystems [15–17]. However, from an engineering point of view, aquatic vegetation practices have not been frequently encouraged because of increased flow resistance, sediment transport effects

and decreased flood discharge efficiency compared to unvegetated regions [18]. Thus, a better understanding of the physical processes governing flow resistances in vegetated areas could help resolve these conflicting engineering and ecological considerations [19].

The studies conducted so far in this field consider straight flumes with different types of vegetation on the bed. Nepf [20] developed a physically based model to predict turbulence intensity and diffusion within rigid and emergent vegetation. Ghisalberti and Nepf [21] emphasized the importance of vegetation in open channels and used a model of vegetation to study the flow features in a straight flume. This model showed that the coherent vortices of vegetated shear layer dominate the vertical transport. Since then, many laboratory and numerical studies have been conducted to discuss the effects of vegetation on moving fluids [22–25]. For example, Nezu and Sanjou [26] investigated the turbulence characteristics and the evolution of coherent structures in a flume with rigid and submerged vegetation. They highlighted that the development of sweep and/or ejection events near the top of vegetation affects the exchange processes between vegetation and the overflow. Zhao and Cheng [27] used an array of rigid cylindrical rods to simulate emergent vegetation stems that were subject to unidirectional open channel flows. Until now, straight open channel flows with vegetation have received enough research attention. To address the new objectives of river restoration and environmental flood management, a better understanding of hydraulic features of flow through local non-submerged rigid vegetation in Y-Shaped confluences channel is required.

In this study, we focus on how local non-submerged vegetation influence hydraulic features in a Y-shaped confluence channel. A physical model was used to simulate changes of flow state, velocity distribution, turbulence structures, and turbulent kinetic energy caused by vegetation by comparing the results with non-vegetated conditions under the same flow regime. The primary objective is to investigate the features of distribution of velocity, secondary flow structure and turbulent kinetic energy with the existence of vegetation in Y-shaped confluence channel.

## 2. Laboratory Experiments

### 2.1. Experimental Setup and Measurement Technique

Experiments were conducted in a flat-bottom Plexiglas flume (Figure 1) in the Hydraulics Laboratory of Hohai University. Water was pumped into a stilling cistern and then flowed into the flume. The length of the upstream tributaries was 3 m, and the widths were 0.26 and 0.22 m, respectively. The length of the downstream main stream was 6 m, and the width was 0.4 m. In order to make this study have a certain practical significance, the design of the flume model was combined with the analysis of the morphological characteristics of the Xitiaoxi River Basin (30°23′–31°11′ N, 119°14′–120°29′ E) to select the parameters (Table 1). The average width before and after the intersection of rivers and the intersection angle at the intersection were counted respectively. Based on the analysis of the morphological characteristics of river network and the actual conditions, the convergence angle between the tributaries (the angle between the geometric axes) was 60°. A large number of non-submerged rigid vegetation, such as reeds, existed on both sides of these Y-shaped intersections. Polyvinyl chloride(PVC) baseboards ($1 \times 0.1 \times 0.01$ m) were used to cover the entire bottom of the flume. Rigid cylinders were used to simulate rigid vegetation (8 mm diameter, 0.2 m height). The flexural rigidity of simulated rigid vegetation was calculated using the approach of Łoboda et al. [28], based on Niklas [29] and ASTM [30]. The flexural rigidity of simulated rigid vegetation is 40,960 N· mm$^2$.

The distance measured between the plants (rigid cylinders) was 0.025 m, and the linear spacing was 0.1 m. Vegetation was distributed on both sides of the flume in 2-m-long bands along the two tributaries, and 5-m-long bands along the main stream were planted perpendicularly with the artificial vegetation. In addition, two transition segments and a tailgate were installed to prevent large-scale disturbances from the inlet, thus enabling the development of a quasi-constant water flow by depth.

The outlet and the inlet, which were both connected to the tank, enabled continuous recirculation of the steady-state discharges.

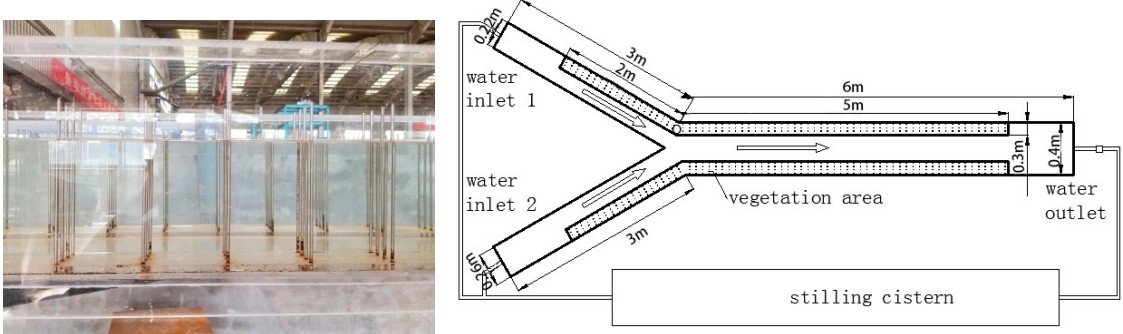

**Figure 1.** The sketch of the experiment.

**Table 1.** The morphological characteristics of the Xitiaoxi River Basin.

| Y-Shaped Confluence Channel | Average Width of Left River before Confluence (m) | Average Width of Right River before Confluence (m) | Average Width of River after Confluence (m) | The Convergence Angle (°) |
|---|---|---|---|---|
| 1 | 100.21 | 107.81 | 198.81 | 58.59 |
| 2 | 69.06 | 80.15 | 106.15 | 48.35 |
| 3 | 27.81 | 30.83 | 54.65 | 30.81 |
| 4 | 98.23 | 100.88 | 163.15 | 60.98 |
| 5 | 54.14 | 73.18 | 100.88 | 69.27 |
| 6 | 33.31 | 48.02 | 54.15 | 75.025 |
| 7 | 32.61 | 38.17 | 44.63 | 92.73 |
| 8 | 29.56 | 40.3 | 74.14 | 45.57 |
| Average | 55.616 | 64.918 | 99.570 | 60.166 |

This experiment used two flowmeters to measure the discharge of both tributaries. 3D velocities and velocity fluctuations were measured using a 3D sideways-looking acoustic Doppler velocimeter (ADV) manufactured by SonTek, Inc. (San Diego, CA, USA). The ADV technology is based on the pulse-to-pulse coherent measurement method. The instrument consists of three modules: a measuring probe, a conditioning module, and a processing module [31]. In this study, the ADV was used to sample each measurement point at a frequency of 50 Hz for 30 s. The WinADV-program, a post-processing program, was used to filter and post-process the sampled data. Data with average correlations less than or equal to 80% were filtered out. The result was that each point had a total of 1500 instantaneous data points, which ensured the adequacy and accuracy of the dataset. To achieve three-dimensional movement control of the ADV, a special regulating device was used to realize three-dimensional free positioning movement of the ADV probe in the test flume. The probe could be easily moved between measurement lines and sections. The ADV was mounted in a wood frame across the center section of the test segment and could be easily moved upstream or downstream, so that all sampling points were vertically aligned. The probe could also channel real-time data to the user's computer through a data acquisition program.

*2.2. Test Series Description*

To conduct comparative experiments and analyses, both the flume with vegetation and one without vegetation were measured while keeping other conditions unchanged. In this experiment, 18 sections were set up (two sections each on the tributaries and 14 sections on the main stream after the intersection). The coordinate origin was set at the bottom of the flume, directly below the intersection point of the two tributaries.

Before the intersection, five perpendiculars were established on each section of the tributaries, with each perpendicular having eight measurement points. In other words, there were 40 points in each section. After the intersection, eight perpendiculars were established on each section of the main stream, with each perpendicular having eight measurement points. This means that there were 64 points for each section. There were five survey lines on the two tributaries and seven survey lines on the mainstream. Figure 2a,b show the sections and the measuring lines.

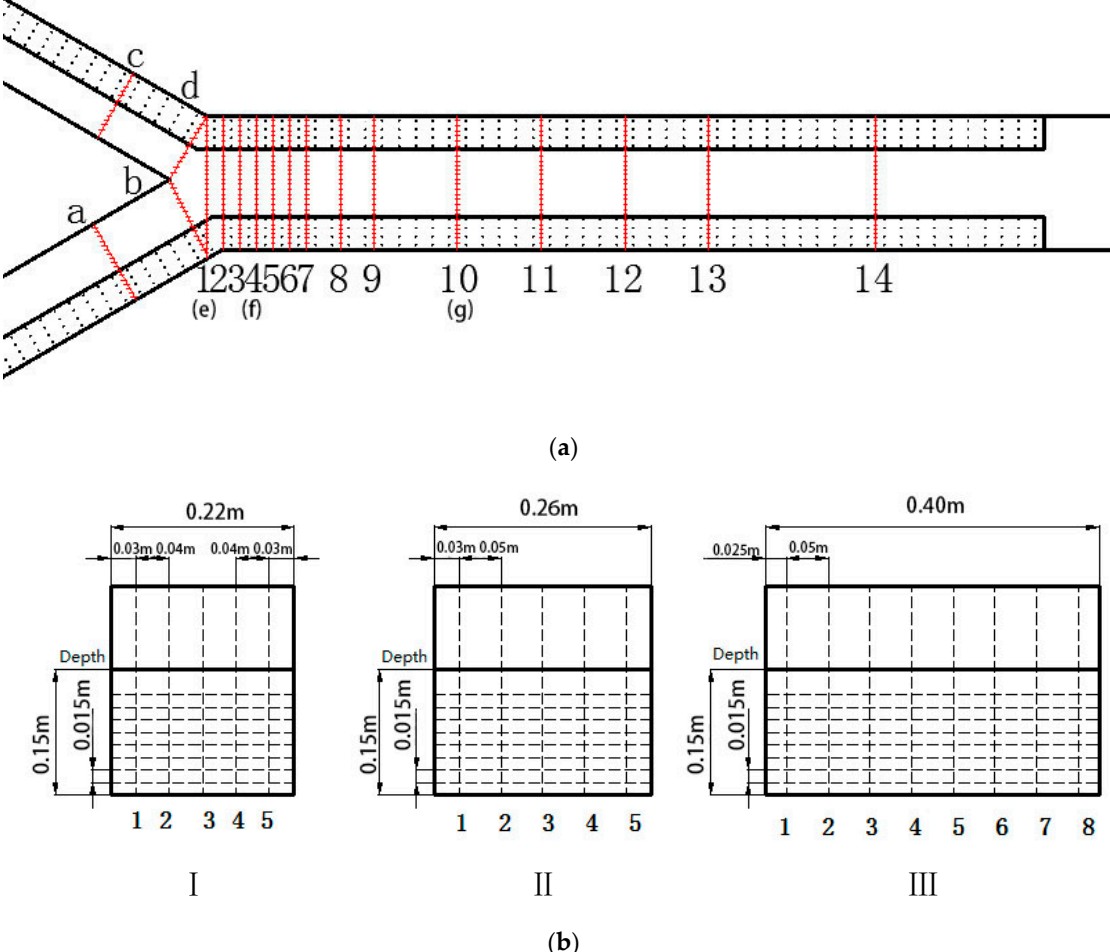

(a)

(b)

**Figure 2.** The sections and measuring lines; (**a**) Distribution of measuring cross sections; (**b**) Distribution of measuring lines.

The 3-D Doppler ultrasound anemometer was used to measure the flow velocities on 18 typical cross-sections. By comparing date, seven of those typical cross-sections were chosen to study. The seven sections were the initial section and intersection of both tributaries and sections 1, 4, and 10 of the main streams, as seen in Figure 2a. These seven cross sections were defined as cross Section a, cross Section b, cross Section c, cross Section d, cross Section e, cross Section f, and cross Section g. Section a is 2 m upstream of the right confluence apex, Section b is the right confluence apex, Section c is 2 m upstream of the left confluence apex, Section d is the left confluence apex, Sections e, f, g are 0, 0.2, 0.6 m downstream of the confluence apex, respectively. Vertical measuring lines (Figure 2b) were arranged on each cross-section. All the experimental data were derived from detailed high-resolution measurements on each measuring line.

In the flow characteristics test, the flow rate of the left tributary was $Q_1 = 20$ m³/h, and the flow rate of the right tributary was $Q_2 = 20$ m³/h. Therefore, the confluence ratio = $Q_1/Q_2 = 1.0$, and the intersection angle = 60°. Some parameters of the experiment are shown in Table 2, Q denotes to the

flow rate; b and h means the width and water section depth, respectively; V points to the velocity of the water section; $R_e$ and $F_r$ is the Reynolds number and Froude number of the flow, respectively.

**Table 2.** Parameters of the experiment.

| Element | $Q$ (m$^3$/h) | b (m) | h (m) | V(m/s) | $R_e$ (10 °C) | $F_r$ |
|---|---|---|---|---|---|---|
| Left tributary | 20 | 0.22 | 0.15 | 0.17 | 8168 | 0.14 |
| Right tributary | 20 | 0.26 | 0.15 | 0.14 | 7584 | 0.12 |
| Main stream | 40 | 0.4 | 0.15 | 0.19 | 12135 | 0.16 |

## 3. Analytical Methods

### 3.1. Velocity Distribution

Li et al. [32] proposed a method to calculate the velocity of a curved open channel with partially non-submerged rigid vegetation by comparing the integral of the longitudinal depth averaged velocity between vegetated and non-vegetated conditions to analyze the changes in the velocity distribution. The streamwise velocities at different water depths in different sections under vegetated conditions were depth averaged along the depth to obtain the corresponding average streamwise velocity. Due to the limited number of measurement points, the area method of experimental points can be used to circumscribe the area and to calculate the experimental measurement points. By comparing the integral of the streamwise depth velocity between vegetated and non-vegetated conditions, the changes in velocity distribution could be analyzed.

### 3.2. Secondary Flow Structure

Secondary motions are commonly present in open channel flows. In contrast to the facility in identifying longitudinal bedforms, secondary flows are much more difficult to measure because of their relatively small magnitude [33]. Around the 1950s, the existence of cellular secondary flows in rivers was only inferred from laterally periodic distribution of primary flow and sediment concentration, rather than being confirm through direct velocity measurements, by geologists and river engineers [34]. The precise filed measurement of cellular secondary flows was possible only after the ascent of the electromagnetic current meter (EM) and acoustic Doppler velocimeter (ADV) [35]. The water mixed energetically in the Y-shaped channels, and three-dimensional flow characteristics were prominent. A single vector or contour map was not enough to explain these complex hydrodynamic phenomena. For ease of explanation, this paper provides and explains a longitudinal velocity contour slice map. Each specific section of the diagram is presented and explained. The study presents a comparative analysis of the flow characteristics of the Y-shaped channels with and without vegetation. The velocity data obtained through the experiments were processed using the Tecplot software.

### 3.3. Turbulent Kinetic Energy

The quantitative analysis of turbulent flow is based on measurements of velocity fluctuations at a single point with local non-submerged rigid vegetation and without any vegetation in a Y-shaped confluence channel. In this study, the mean flow velocity components ($u$, $v$, and $w$) and the velocity fluctuation components in turbulent flow ($u'$, $v'$, and $w'$) correspond to the streamwise, lateral, and vertical directions, respectively. Velocity fluctuations can be defined as deviations from the mean velocity. The turbulent kinetic energy was calculated as noise free [36].

In general, turbulent kinetic energy can be considered a measure of turbulence intensity [37]. In this study, the turbulent kinetic energy $E_k$ was calculated using Equation (1), which was proposed by Li [38] and defines $E_k$ as follows in terms of streamwise, lateral, and vertical directions:

$$E_k = \rho\left(\overline{u'}^2 + \overline{v'}^2 + \overline{w'}^2\right)/2 \tag{1}$$

where $\rho$ is the density of water and $u'$, $v'$, and $w'$ are the velocity fluctuations of streamwise, lateral, and vertical flow, respectively, and the overbars denote mean values.

## 4. Results

### 4.1. Distribution of Stream-Wise Velocity

The depth-averaged streamwise velocity distribution of the seven typical cross sections (Figure 3) showed that under the retarding effect of vegetation, the depth-averaged streamwise velocities ($U_{avg}$) in the vegetated region were much lower than in the non-vegetated region. A large velocity gradient was apparent near the junction of the vegetated and non-vegetated areas, we can see all the figures show a large velocity gradient was apparent near the junction of the veg and non-veg areas, it not only can be seen from the tributaries, but also from the mainstreams, which showed typical values as reported by Huai et al. [39].

There was a difference in the streamwise velocity distribution between non-vegetated and vegetated cases (Figure 3). The velocities in the vegetated case have a higher gradient than in the non-vegetated case, which can be ascribed to the fact that the retardation caused by vegetation makes the transverse distribution slightly more non-uniform. Along the streamwise direction, the velocities in the vegetated area decreased and those in the non-vegetated area increased when heading downstream, which is consistent with results for open channel flows found by Nikora [40] and Caroppi [41].

Three-dimensional flow characteristics were prominent after the intersection. The depth-averaged streamwise velocity distribution was not enough to explain the complex hydrodynamic phenomena. Clearly, the flow rate gradient was larger with vegetation than without vegetation, which indicates that the presence of vegetation caused drastic changes in the flow structure of the river (Figure 4). With vegetation (Figure 4a), the flow velocity in vegetated zones was smaller than in non-vegetated zones. It can also be observed that in the absence of vegetation (Figure 4b), the streamwise velocity distribution was polarized in the longitudinal direction, with the branch on the left side going downstream having a large flow area, whereas the area close to the right-side tributaries near the bottom was in a low-velocity zone. Downstream of the confluence in non-vegetated cases, a small velocity separation zone appeared on the lateral surface of the left tributary. This is consistent with Wang's results [42]. After vegetation planting (Figure 4a), the polarized character of the streamwise velocity distribution in the longitudinal direction disappeared, the downstream flow velocities in the vegetated areas in the left and right branches obviously diminished, and the flow velocity distribution with depth became insignificant.

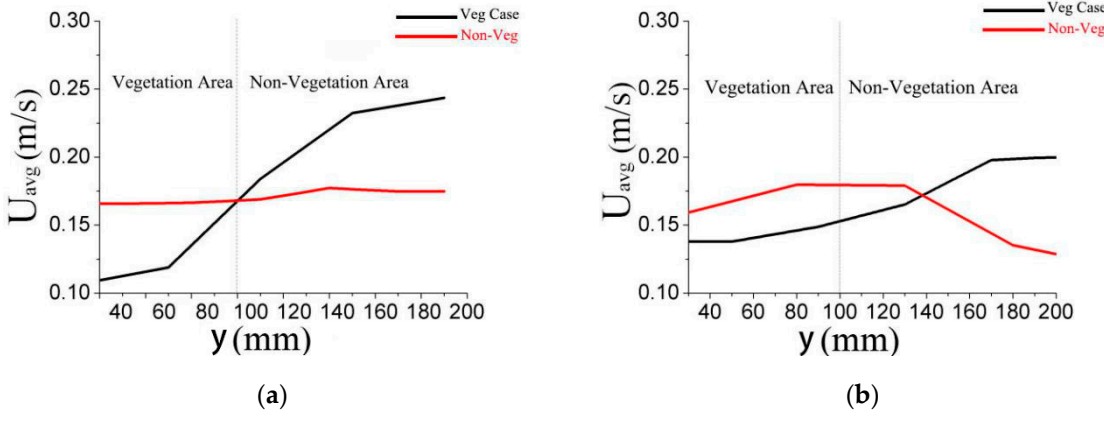

**Figure 3.** *Cont.*

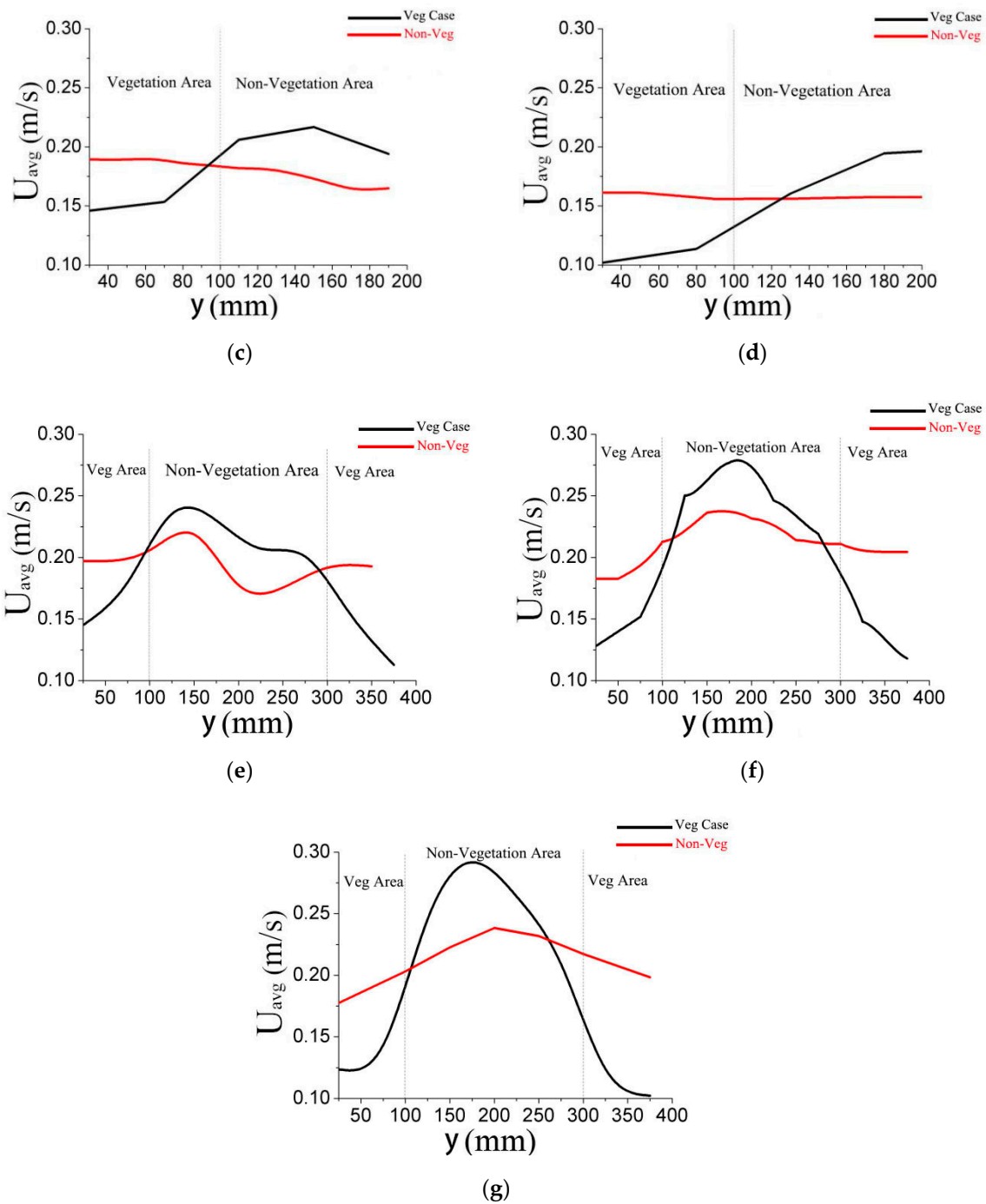

**Figure 3.** Comparison of depth-averaged streamwise velocity transverse distribution between vegetated and non-vegetated cases, where $U_{avg}$ is the depth-averaged streamwise velocity and y is the flume width. (**a**) Cross Section a; (**b**) Cross Section b; (**c**) Cross Section c; (**d**) Cross Section d; (**e**) Cross Section e; (**f**) Cross Section f; (**g**) Cross Section g.

*4.2. Secondary Flow Structure*

There was a difference in Secondary Flow Structure by comparing the experimental results for the confluence of the Y-shaped channels with vegetation (Figure 5) and without vegetation (Figure 6). For the mainstream Sections (e), (f), and (g), when vegetation was present, the velocity difference in the confluence was more remarkable. The Section e is the confluence cross section, without vegetation (Figure 5), the left velocity was higher, and the right velocity was lower. A small clockwise circulation

appeared at the bottom of the left side. But there was no circulation in the vegetated case (Figure 6). In addition, with the presence of vegetation, the velocity rapidly became partitioned, with the velocity on both sides decreasing with resistance from vegetation and increasing in the middle. Cross sections e and g showed that the higher-velocity area extended to the center in the non-vegetated case, the bottom of the separation zone became larger, the secondary flow phenomena along the left bank slowly disappeared, and a slight counterclockwise motion appeared in the lower right corner in the mainstream cross Section f. This result is similar to the report by Vaghefi et al. [43]. Under the influence of vegetation resistance, the low-velocity zone gradually became wider, the high-speed zone gradually narrowed, the low-speed area on the left changed faster than the low-speed area on the right, and there was no circulation.

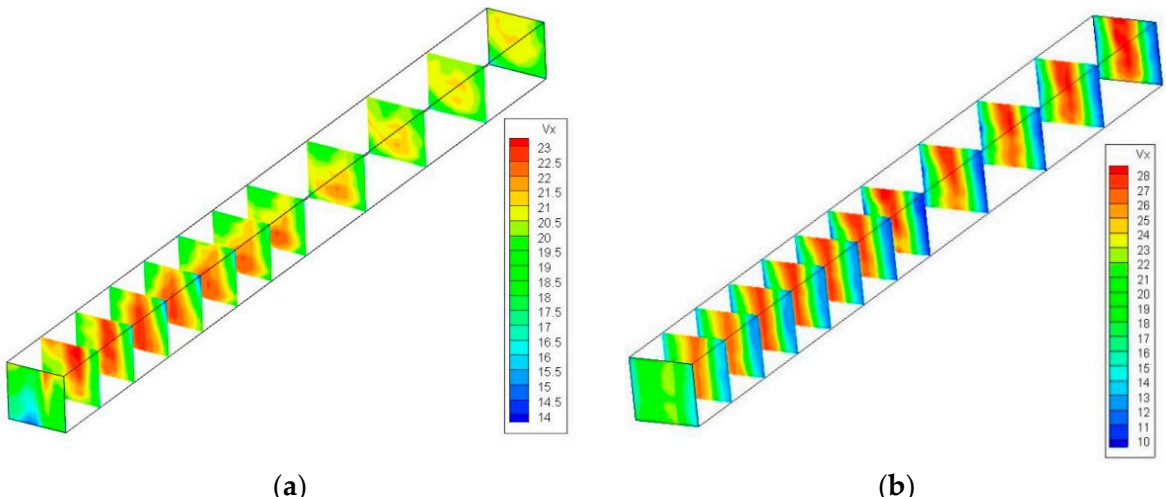

(**a**)                                                                                                 (**b**)

**Figure 4.** Slice diagram of streamwise velocity contour after intersection: (**a**) vegetated case; (**b**) non-vegetated case.

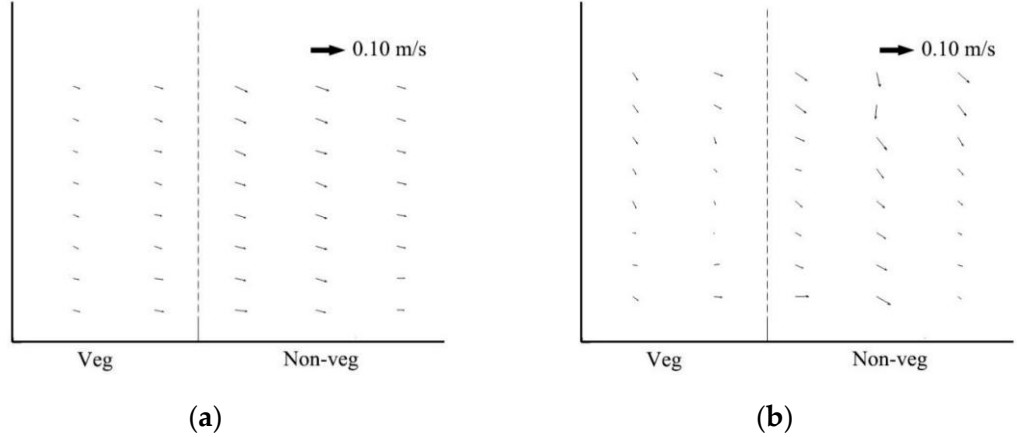

(**a**)                                                                                                 (**b**)

**Figure 5.** *Cont.*

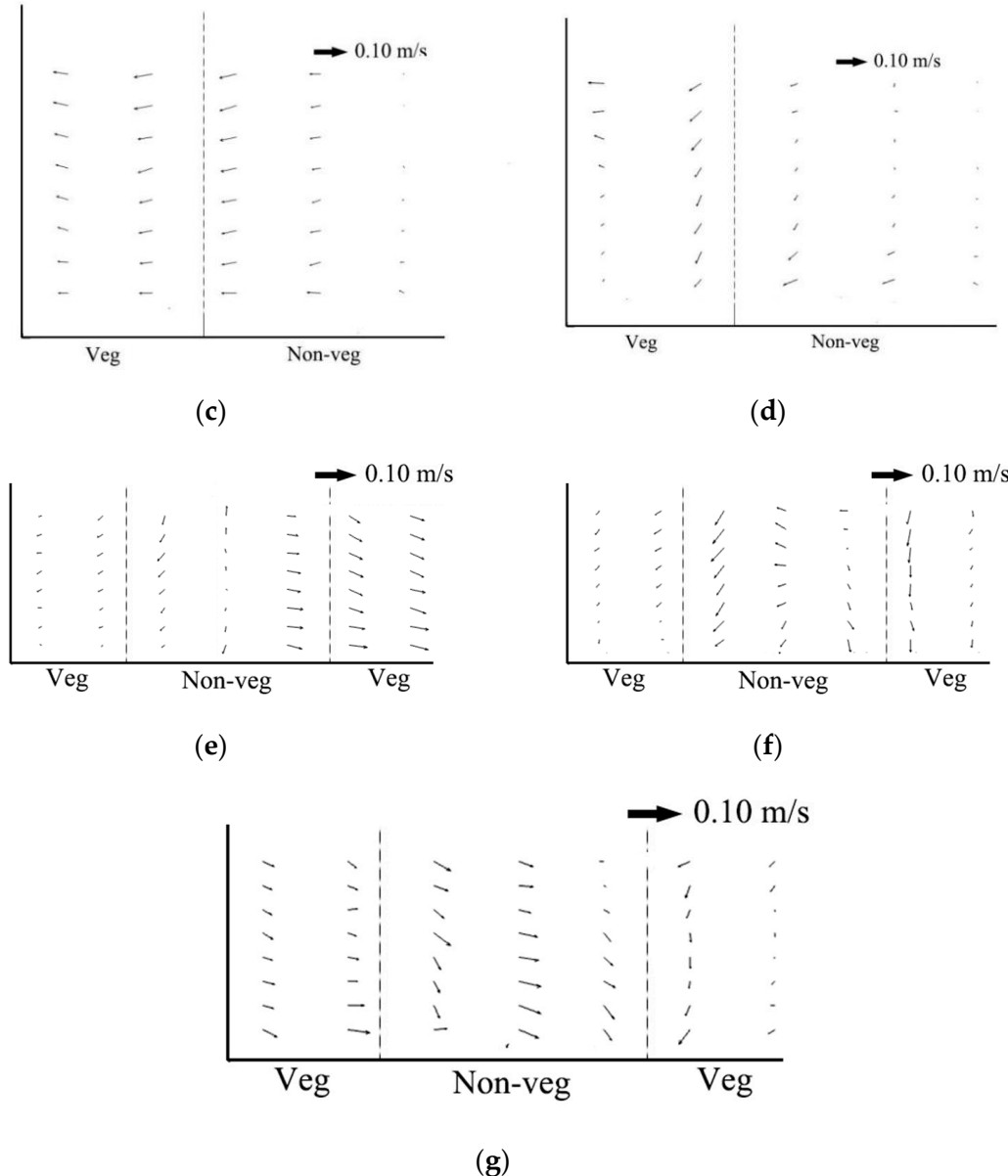

**Figure 5.** Comparison of secondary flow structures with vegetation. (**a**) Cross Section a; (**b**) Cross Section b; (**c**) Cross Section c; (**d**) Cross Section d; (**e**) Cross Section e; (**f**) Cross Section f; (**g**) Cross Section g.

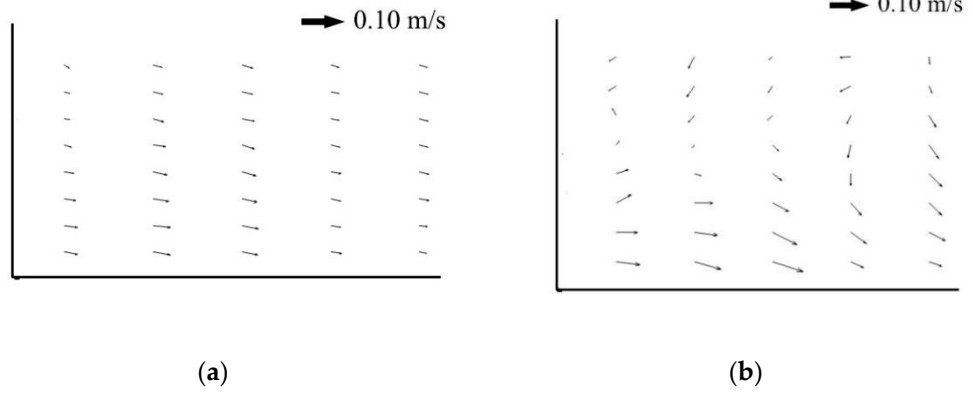

**Figure 6.** *Cont.*

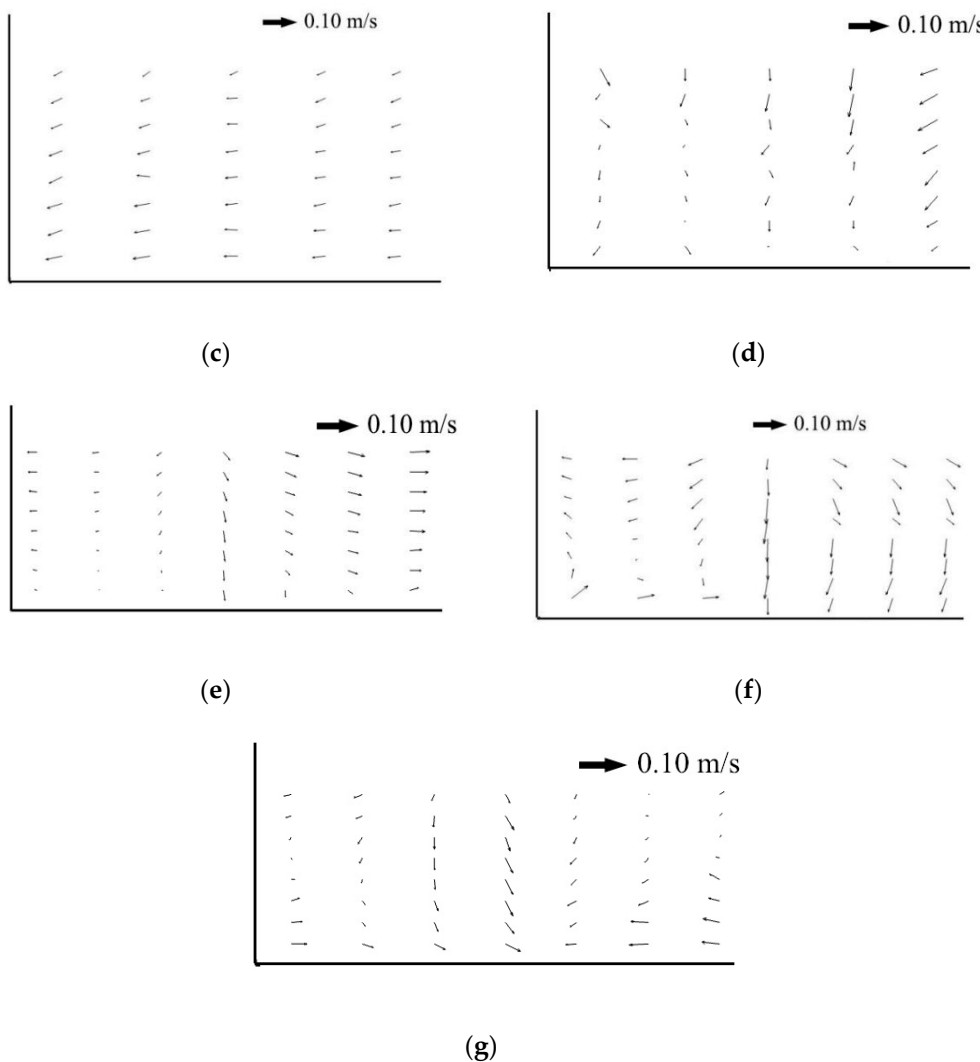

**Figure 6.** Comparison of secondary flow structures without vegetation. (**a**) Cross Section a; (**b**) Cross Section b; (**c**) Cross Section c; (**d**) Cross Section d; (**e**) Cross Section e; (**f**) Cross Section f; (**g**) Cross Section g.

*4.3. Turbulent Kinetic Energy*

The relational graphs of the change of turbulent kinetic energy with depth were shown in Figures 7 and 8, respectively. According to Figures 7 and 8, the results showed that the turbulent kinetic energy in the tributaries was much larger with vegetation than without it. The maximum value with vegetation was greater than 20 Pa (Figure 7a–d), compared to a maximum of less than 15 Pa (Figure 8a–d) without vegetation. The turbulent kinetic energy of the vegetated area was less than that of the non-vegetated area, and the turbulent kinetic energy showed a remarkable increase near the junction of the vegetated and non-vegetated areas. Although the numerical values of turbulent kinetic energy for each section at different depths were different, the distribution trends in the tributaries were consistent. After convergence, as shown in Figure 7e–g, the turbulent kinetic energy of the non-vegetated area was smaller than that of the vegetated area.

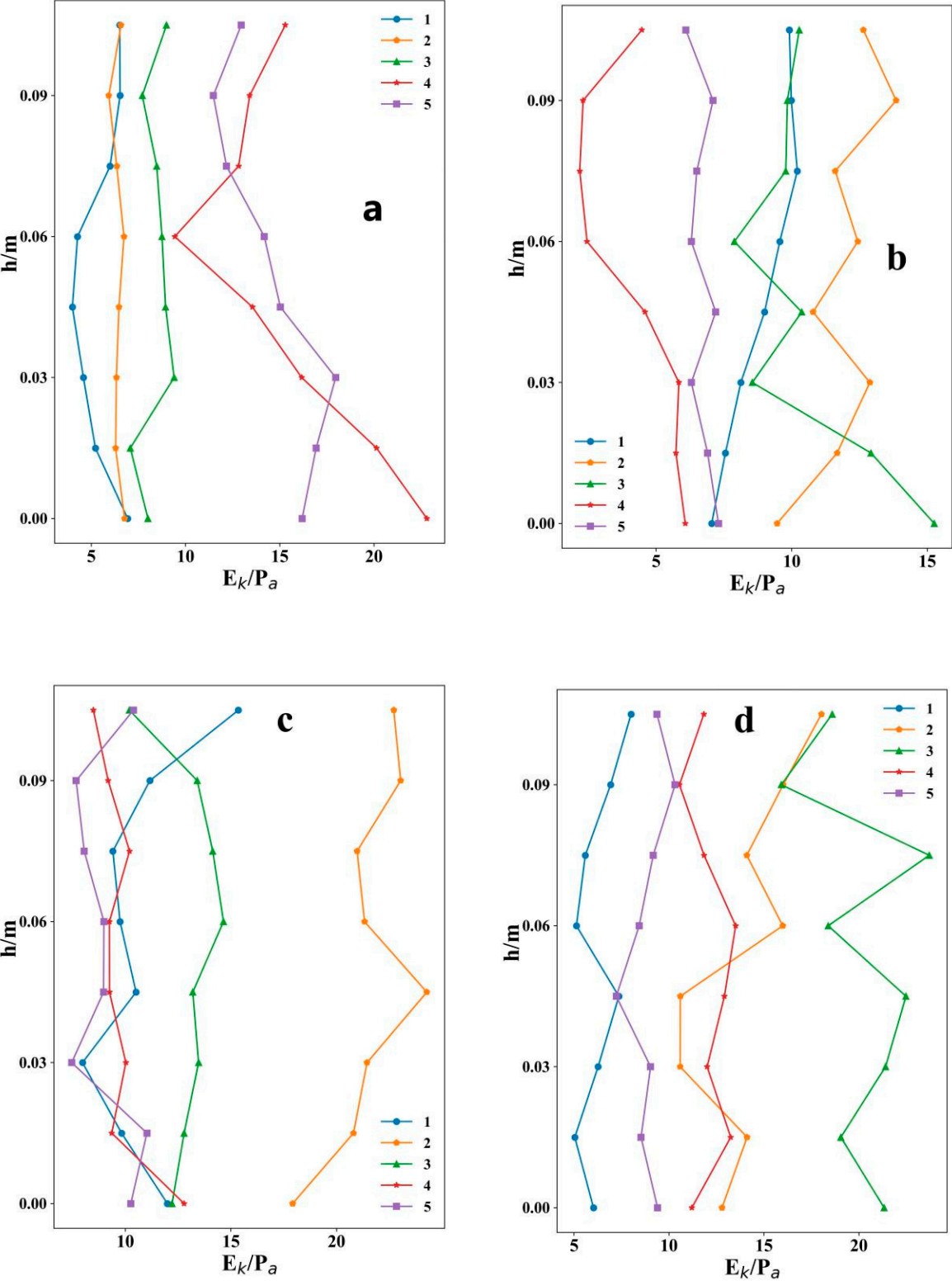

**Figure 7.** *Cont.*

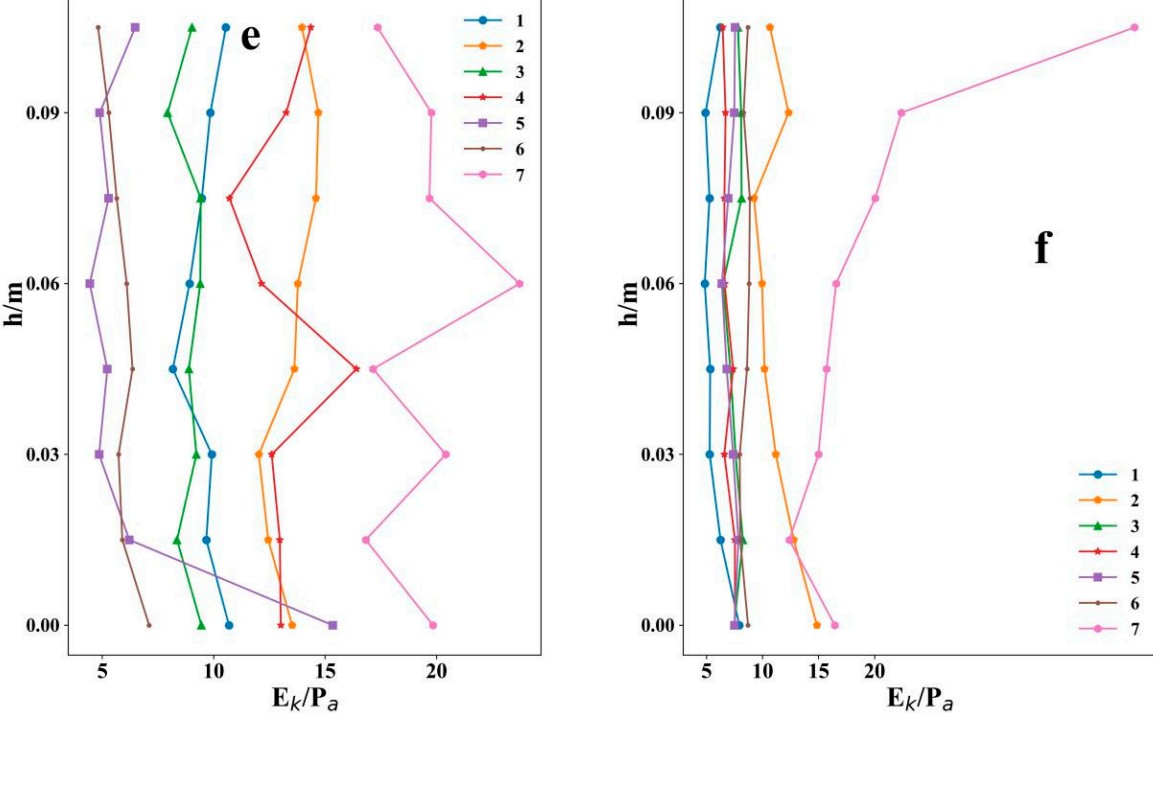

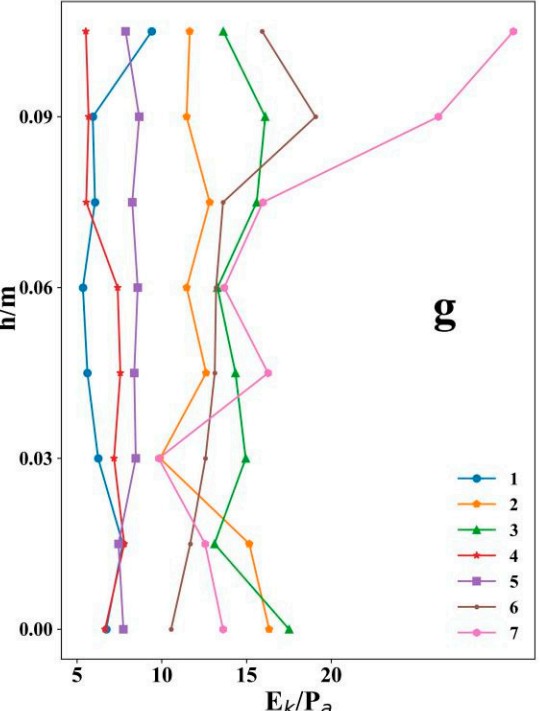

**Figure 7.** Turbulent kinetic energy ($E_k$/Pa); (**a**) Cross Section a at vegetation cases; (**b**) Cross Section b at vegetation cases; (**c**) Cross Section c at vegetation cases; (**d**) Cross Section d at vegetation cases; (**e**) Cross section e at vegetation cases; (**f**) Cross section f at vegetation cases; (**g**) Cross section g at vegetation cases.

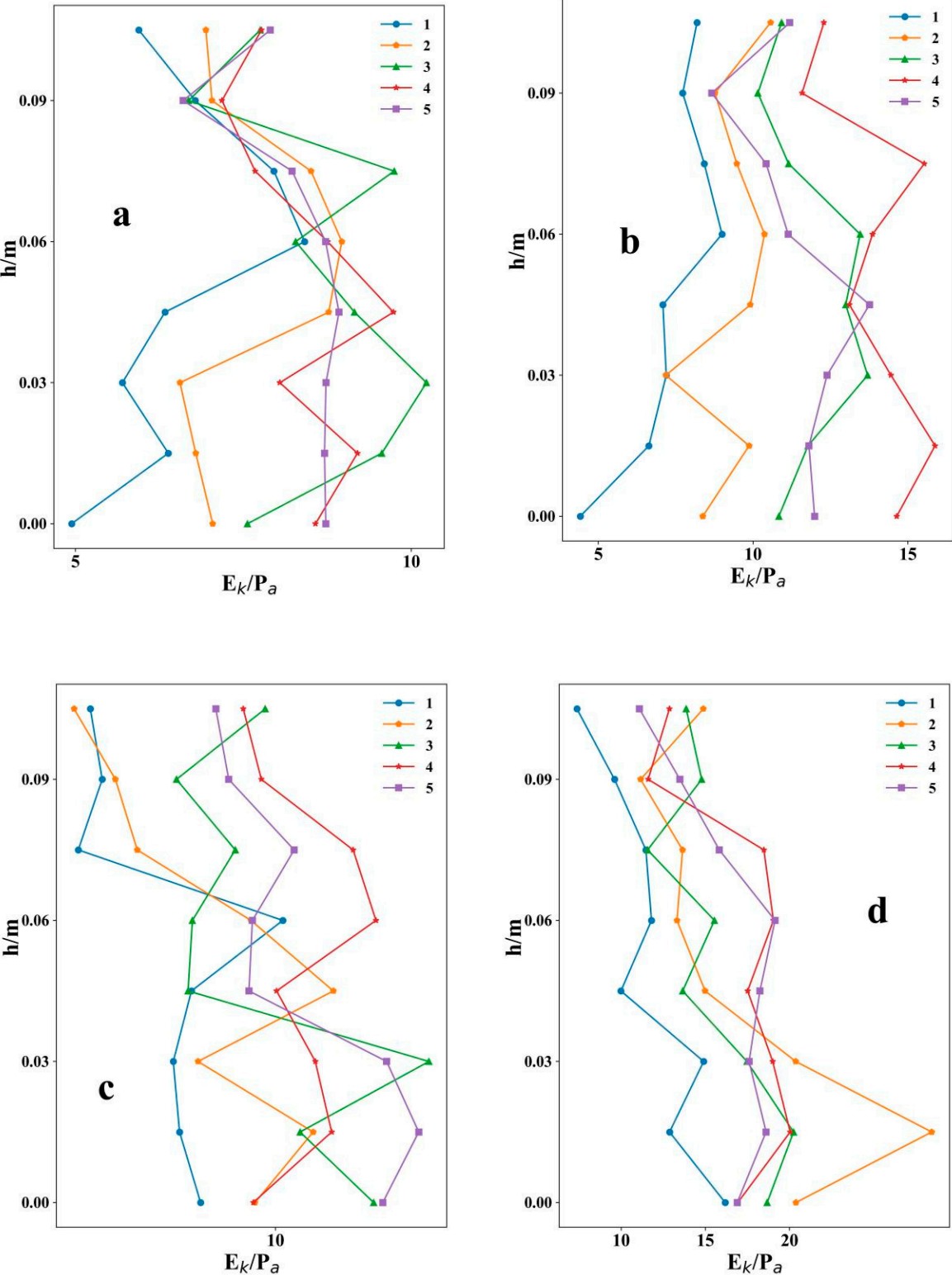

**Figure 8.** *Cont.*

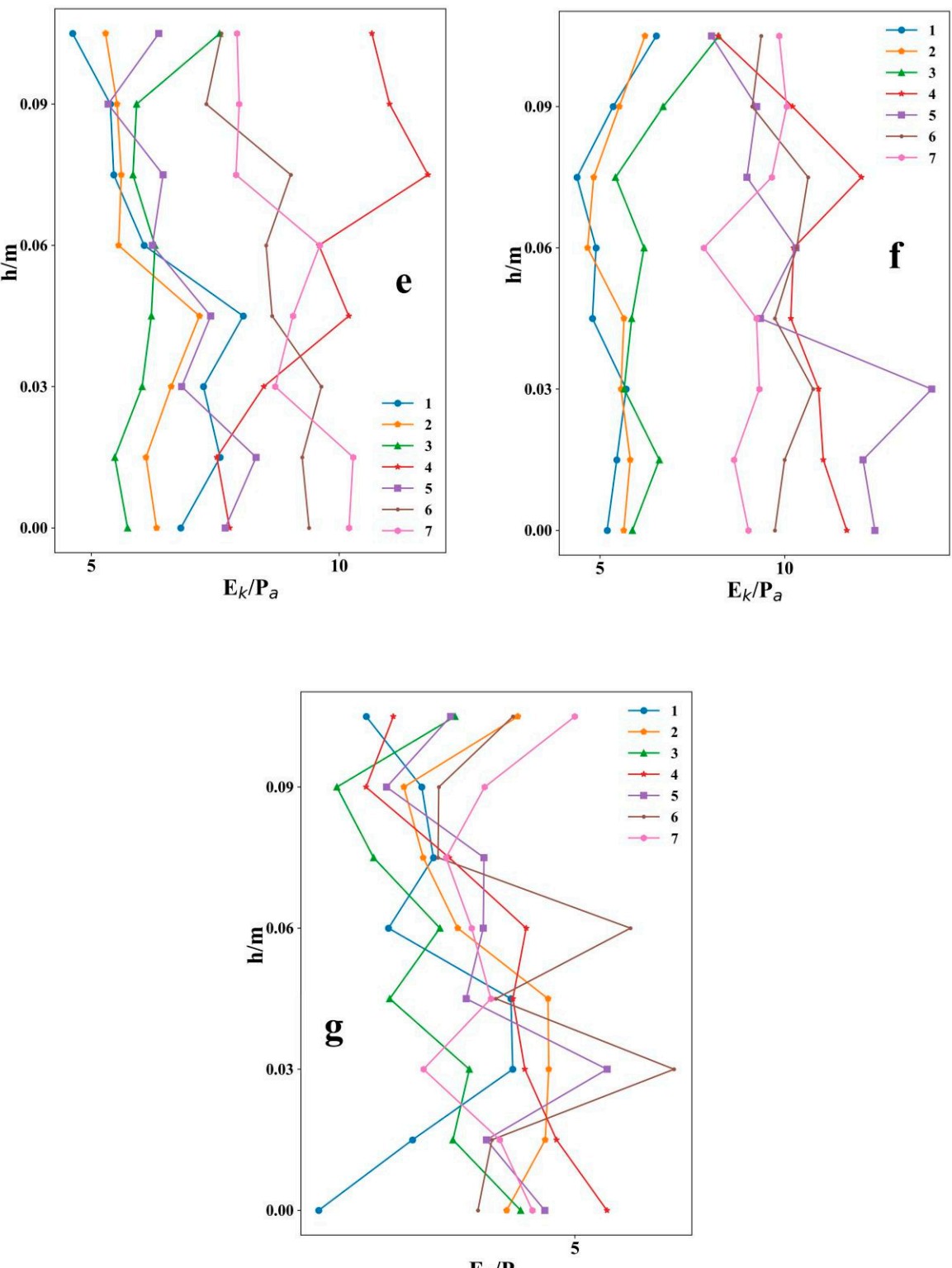

**Figure 8.** Turbulent kinetic energy ($E_k$/Pa); versus relative depth (h/m) with vegetation, versus relative depth (h/m) without vegetation. (**a**) Cross Section a at non-veg cases; (**b**) Cross Section b at non-veg cases; (**c**) Cross Section c at no-veg cases; (**d**) Cross Section d at no-veg cases; (**e**) Cross Section e at no-veg cases; (**f**) Cross Section f at no-veg cases; (**g**) Cross Section g at no-veg cases.

## 5. Discussion

In this study, with the effect of vegetation, the streamwise velocities in the vegetated area were much lower than in the non-vegetated area (Figures 3 and 4), which is consistent with the field study conducted by Chambers et al. [44]. They showed that in natural rivers with slower currents and therefore in the vegetated region, the water current was expected to be lower than in the main stream. The velocities in the vegetated case have a higher gradient than in the non-vegetated case, which can be ascribed to the fact that the retardation caused by vegetation makes the transverse distribution slightly more non-uniform, consistent with the conclusion drawn in the study by Mossa et al. [45]. Along the streamwise direction, the velocities in the vegetated area decreased and those in the non-vegetated area increased when heading downstream (Figures 3 and 4). The reason for this may have been that the main circulation in the Y-shaped confluence changed the internal flow structures, and the velocity distribution varied accordingly, which is consistent with the results from Wang et al. [42]. The vegetation prevented vertical mixing of fluid, and because of resistance, the velocity near the vegetated side was significantly lowered.

A large velocity gradient was apparent near the junction of the vegetated and non-vegetated areas. We can see that all the figures show that a large velocity gradient was apparent near the junction of the veg and non-veg areas, indicating there are must be remarkable mass and momentum exchanges at the junction of these two areas. The secondary flow is occurred for the momentum exchange [46]. The experimental results for the confluence of the Y-shaped channels with vegetation (Figure 5) and without vegetation (Figure 6) were compared. In the vegetation area, no clear circulation is found in the whole area.

The presence of vegetation caused a great change in the internal flow structure and made the flow in non-vegetated areas more intense. When water flows through vegetation, the flow resistance increases and the corresponding velocity decreases. At the same time, the flow will generate turbulence, consuming the flow energy. The turbulent kinetic energy of the non-vegetated area was smaller than that of the vegetated area. This is similar to Liu et al.'s results [47].

Hydraulic properties of riverine vegetation have been widely researched. Many laboratory, numerical, analytical and field studies have been conducted to account for the complex dynamic interactions between vegetation and moving fluids. The channel characteristics of natural rivers are seen to constitute an interdependent system which can be described by a series of graphs having simple geometric forms [48]. There are many Y-shaped confluence channels in natural river networks. This study can be helpful in providing some physiographic implications for practical use.

## 6. Conclusion and Future Works

Due to the presence of vegetation, the flow velocities in the Y-shaped confluence channel were redistributed. Velocities in the vegetation area were much smaller than those in non-vegetation area due to the presence of vegetation. Compared with the flow without vegetation, the streamwise velocity in the vegetated area was much lower than in the non-vegetated area because of the effect of vegetation. A large velocity gradient is generated between the vegetated and non-vegetated areas, indicating a remarkable mass and momentum exchange at the junction of these areas. Under the combined effect of vegetation and the Y-shaped confluence, the streamwise velocities in vegetated areas were much lower than those in non-vegetated areas, and the streamwise velocity of non-vegetated areas in the mainstream in the case with vegetation increased significantly compared with the case without vegetation.

In the tributaries, due to the presence of vegetation, the high-velocity area moved rapidly to the middle of the channel in non-vegetated areas, and the secondary flow phenomenon disappeared. In the mainstream, when vegetation was introduced, circulation disappeared, and the degree of lateral mixing decreased.

The presence of vegetation brought about great changes in internal flow structure. It caused the flow in non-vegetated areas to become more intense, and the turbulent kinetic energy of the tributaries in the cases with vegetation was significantly lower than in the cases without vegetation.

At the junction of vegetated and non-vegetated areas, the turbulent energy increased significantly. The turbulent energy of the non-vegetated areas was significantly greater than in the same position in the absence of vegetation.

This work focused on a single type of aquatic plant in the Y-shaped confluence channel. However, rivers actually include several types of plants. Further studies could investigate different vegetation types and different discharges in the Y-shaped channel. This study focused on flume experiments, but further investigations could combine this approach with numerical modeling.

**Author Contributions:** Conceptualization, X.T. and X.L.; Methodology, T.Y.; Software, R.L.; Validation, J.L., Z.W. and X.T.; Formal Analysis, T.Y.; Investigation, Z.W.; Resources, Z.W.; Data Curation, T.Y.; Writing-Original Draft Preparation, X.T.; Writing-Review & Editing, X.L.; Visualization, X.L.; Supervision, Z.H. and X.L.; Project Administration, X.L.; Funding Acquisition, X.L.

**Funding:** This research was funded by the National Natural Science Foundation of China (NSFC) (Grant No. 51479064, No. 51739002, No. 51479010, No. 51779016), the National College Students' Innovative Training Project China (Grant No. 201710294031), the National Key Project R&D of China (2016YFC0401702), Jiangsu Province Discipline Construction Funded Projects, Project Funded by the Priority Academic Program Development of Jiangsu Higher Education Institutions (PAPD) and PPZY2015A051.

**Acknowledgments:** We are grateful to the Hydraulic Laboratory, Hohai University, for the use of laboratory facilities and to Xu Jingzhao and Mei Shengchen for their thoughtful comments on measurements during the laboratory research.

**Conflicts of Interest:** The authors declare no conflict of interest.

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
