# Peer review of "Hydraulic Features of Flow through Local Non-Submerged Rigid Vegetation in the Y-Shaped Confluence Channel"

_water, doi:10.3390/w11010146_

Reviewer 1 Report

This paper was well written, but will require minor to moderate revisions to improve it before publication. In particular, a more appropriate introduction is required that ties together literature with the context of the paper. Some more details about methodology and how the setup links to the real world should be provided. I have provided a few comments on improving the presentation of results. The paper needs a discussion to contextualize the results and also draw implications to the real world. 

Author Response

Point 1: This paper was well written, but will require minor to moderate revisions to improve it before publication. In particular, a more appropriate introduction is required that ties together literature with the context of the paper. Some more details about methodology and how the setup links to the real world should be provided. I have provided a few comments on improving the presentation of results. The paper needs a discussion to contextualize the results and also draw implications to the real world. 

Response 1: Thanks for your suggestions. According to your suggestions, we reworte the section of Introduction. We re-read a lot of literature related to confluence channel, vegetated flow. Regarding the methodology and how setup links to the real world, in fact,  In order to make this study have a certain practical significance, the design of the flume model is combined with the analysis of the morphological characteristics of the Xitiaoxi River Basin  to select the parameters. The average width before and after the intersection of rivers and the intersection angle at the intersection are counted respectively. Based on the analysis of the morphological characteristics of river network and the actual conditions, the convergence angle between the tributaries (the angle between the geometric axes) was 60°.  We have added this information into methodology section. In addition, we have separated the RESULTS. and DISSCUSSION section into two different chapters and  connected our results to real world and make some comparisons with other researchers. Thanks again for reviewing our paper!

Reviewer 2 Report

This is an interesting paper with nice interesting data.  However, it is lacking in analysis.  

There are a number of fairly simple things that could be done in terms of understanding the data in the analysis section.  The planted and unplanted sections could be treated as pipes in parallel.  This is a closed pipe assumption, but the analysis would be the same just realize that downstream the head loss in both channel sections will be the same and the head loss and velocity must be consistent with this assumption.  This is a rather simple straight forward analysis in which you show that the outcome measured is consistent with the assumptions made and the expectations these assumptions create.

The authors have stated that they are evaluating the structure of the turbulent flow field, and yet then never really discuss turbulent structure in terms that are common in the industry.  Usually when we think about turbulent structure there are a number of parameters that are used to describe the structure.  The Reynolds number of the flow field is assumed, the length scale of the eddies associated with energy input into the system is established, and the Kolmogorov micro scale is estimated.  It is widely accepted that the Kolmogorov micro scale represents the length scale of the turbulent eddy which has a Reynolds number of 1.  That is the inertial and viscous forces are in balance for turbulent eddies of that size.  Above this length scale the region is fully turbulent and below this length scale the turbulent eddies are bound by viscosity and the energy is dissipated as heat.

In a mixed reactor one assumes or deduces a length scale for energy input into the system.  One would expect that changes in the turbulent structure would occur between the energy input length scale and the Kolmogorov micro scale or lower boundary of turbulence length scale. It is assumed that as one goes to length scales smaller then the Kolmogorov microscale, there is less true turbulence and there is a mix of pseudo turbulence and shear fields.  Engineers will also discuss energy in the flow field relative to G or root mean square velocity gradient.  I am not sure it is possible to discuss the structure of the turbulent flow field without some discussion of the large concepts mentioned here.  There are a number of conceptual models for turbulence, the authors need to select one, discuss that model and then explain how they have estimated the impact of the vegetation on turbulent structure. 

Author Response

Thanks for your suggestions! According to your suggestion, we have made a deeper analysis. In the revised version, we have readjusted this article. We have re-read the relevant literature and added more detailed content in the introduction of the method. We separated RESULTS and DISCUSSION, and compared them with previous studies, and combined the actual situation. At the same time, because of limited time, I can not apply all your suggestions to this article, but your comments will be of great help to our future research. Thanks again for reviewing our paper!

Reviewer 3 Report

General comments:

Interesting experiment.

But, the last sentence of the INTRODUCTION says: The results will have a great contribution to the future of eco-hydaulics....ok...Where have you solved this promise?Your results shoulb be ”translated” for scientific community and linked with other results (from laboratory experiments or real world). (in the DISCUSSION section)

The authors should separate (without any doubts) the RESULTS. and DISSCUSSION section into two different chapters.

The results have to be clearly presented and DISSCUSSIONS..have to be extented and linked with real world and others studies as I said already.

The authors should connect their results to real world and make some comparisons....If they do not do that....these interesting results obtained by them, have no purpose.  

This manuscript must never be published until the authors read and cite the paper of Leopold and Maddock, 1953: The hydraulic geometry of stream channels and some physiographic implications. Atfer the authors will read this paper, it is possible to make better use of their own results.

Atfer the authors will solve all these issues, the manuscript can be considered for publication.

Specific comments:

-

Author Response

Thanks for your suggestions! According to your suggestions, We have re-read the relevant literature and added more detailed content in the introduction of the method. We separated RESULTS and DISCUSSION, and compared them with previous studies, and combined the actual situation. Thank you for recommending Leopold and Maddock's book to us. After reading it, we found it is of great help to our research. All basic research should serve the application.Thanks again for reviewing our manuscript! More details were shown in the revised manuscript.

Reviewer 4 Report

The manuscript covers an interesting and clearly understudied topic, modelling of hydrodynamics in the vicinity of vegetation and without vegetation in the Y-shaped confluence channel. However, the manuscript suffers from several shortcomings that need to be addressed before it can be accepted. I address here several major general comments and I have included more specific comments in the attached pdf-file.

Authors should provide a wider background, eg. referring to a larger amount of papers of key scientists in the field of hydrodynamics with the occurrence of vegetation, e.g. Vladimir Nikora, Heidi Nepf, Alex Sukhodolov, In the references, there are only two papers with these names. I recommend, for example, papers of Nikora (2010), Nepf (2012) and Sukhodolov (2015). In 2017, there was published the paper of Sukhodolov et al., which concerns measurements of river confluence. Authors based only on older works, whereas recently there have been published various papers, which may bring something interesting to the manuscript. The literature list covers only one paper from 2017, one from 2015, and the rest of them is older. Relatively recently the paper of Przyborowski et al. (2018) appeared in the Water journal. It concerns measurements with ADV. Maybe, it worth focusing on the methodology and/or results, as the paper also includes measurements with aquatic vegetation.

The English of the manuscript needs to be polished and corrected, especially with tenses of presented results.

The section Results and Discussion must be rewritten, as there is no discussion there. The style of the presentation of the results also must be changed. There is no single discussion with other papers in this section!

Moreover, the authors were very sloppy with the preparation of the manuscript. There are many errors with the references and others (see PDF).

Author Response

Thanks for your suggestions! According to your suggestions, We have re-read the relevant literature and added more detailed content in the introduction of the method. We spent a lot of time reading  papers of Vladimir Nikora, Heidi Nepf, Alex Sukhodolov. After reading them,It is not only of great help to our present research, but also of great help to our future research. We have read every paper you recommended to us, we found they can help us explain something, so we cited some of them. We separated RESULTS and DISCUSSION, and compared them with previous studies, and combined the actual situation. Thanks for reviewing our manuscript.

Round  2

Reviewer 2 Report

The paper as presented is fine.  I think there is significant additional value that can be gotten from this experimental work be applying additional theoretical tools to the data as it currently exists.  Perhaps this is for a later paper.  This paper stands alone and is acceptable.  I would point out two the authors that there is a substantial body of literature that links the velocity gradient information to the turbulent structure theory.  There is another paper in here with a little digging.

Some editorial (word-smithing ) comments

In the following, I have made a few minor editorial changes that I hope do not substantially impact the authors intent and only provide some suggested English changes that may reflect regional use of the language and should not be considered binding for the author.

42 The aquatic vegetation is ubiquitous in natural rivers. The vegetation have many ecological,

43 aesthetic and economic benefits, such as providing terrestrial wildlife habitat; improving water

61overflow. Zhao and Cheng [27] used an array of rigid cylindrical rods to simulate emergent

62vegetation stems that were subject to unidirectional open channel flows. Up to now, straight open

63channel flows with vegetation have received significant research attention. To address the new

64 objectives of river restoration and environmental flood management, a better understanding of

67 In this study, we focus on how local non-submerged vegetation influence hydraulic features

Author Response

Point 1: The paper as presented is fine. I think there is significant additional value that can be gotten from this experimental work be applying additional theoretical tools to the data as it currently exists. Perhaps this is for a later paper. This paper stands alone and is acceptable. I would point out two the authors that there is a substantial body of literature that links the velocity gradient information to the turbulent structure theory. There is another paper in here with a little digging.

Some editorial (word-smithing ) comments: In the following, I have made a few minor editorial changes that I hope do not substantially impact the authors intent and only provide some suggested English changes that may reflect regional use of the language and should not be considered binding for the author.

Response: Thanks for your recognition of our work. We have made some minor editorial changes according to your suggestions. We hope our research will be helpful to river hydraulics in the future. Thanks again for reviewing our manuscript.

Reviewer 4 Report

The paper has been significantly improved since the first time it was reviewed. I appreciated time and effort that authors made during the revision. It seems that they answered almost to all comments. 

In the current form the paper represents better paper, which shows interesting results and should be cited by the readers in the future. However, I suggest to make some corrections, eg. some of them were missing from the previous reviews. 

I also suggest rewrite some sentences as now, they are confusing (e.g., lines 189-190) and change the description of figures in some places, e.g. lines 206-207 or 248-249.

Please check the manuscript to remove mistakes, eg. lne 323. In citation after 'al' should be a dot, e.g. Mossa et al. [42]. Other oversights: you have citation (Figure X), and you also have citation (Fig.x).

In the manuscript, there is still missing information about morphology, shape as well as biomechanics influence on the processes.

Author Response

Point 1: The paper has been significantly improved since the first time it was reviewed. I appreciated time and effort that authors made during the revision. It seems that they answered almost to all comments.   

In the current form the paper represents better paper, which shows interesting results and should be cited by the readers in the future. However, I suggest to make some corrections, eg. some of them were missing from the previous reviews. 

Response: Thanks for your recognition of our work. We did spend a lot of time revising the manuscript according to your suggestions. We added some information about morphology and shape (See lines 80-89). Besides, we analysed the morphological characteristics of river network (see Table 1), then set the convergence angle was 60°.

Point 2: I also suggest rewrite some sentences as now, they are confusing (e.g., lines 189-190) and change the description of figures in some places, e.g. lines 206-207 or 248-249.

Response: Thanks for your comments. We apologized for this confusion and checked the whole manuscript carefully. Then we rewrote some sentences. And we changed the discription of figures in the whole manuscript, eg lines 206-207 and 248-249, according to your comments.

Point 3: Please check the manuscript to remove mistakes, eg. lne 323. In citation after 'al' should be a dot, e.g. Mossa et al. [42]. Other oversights: you have citation (Figure X), and you also have citation (Fig.x).

Response: Thanks for your comments. We checked the whole manuscript carefully, and removed all the mistakes you mentioned, eg. Have the same citation (Figure X).

Point 3: In the manuscript, there is still missing information about morphology, shape as well as biomechanics influence on the processes.

Response: Thanks for your comments. We found some literature about this information (eg. Łoboda, A.M.; Przyborowski, Ł.; Karpiński, M.; Bialik, R.J.; Nikora, V.I. Biomechanical properties of aquatic plants: The effect of test conditions.). We added the information about morphology, shape and biomechanical properties in the latest revision.